# Interventions for Sensory Over-Responsivity in Individuals with Autism Spectrum Disorder: A Narrative Review

**DOI:** 10.3390/children9101584

**Published:** 2022-10-19

**Authors:** Huan-Ling Yuan, Cynthia Y. Y. Lai, Mabel N. K. Wong, Tak Chun Kwong, Yat Sze Choy, Steve W. Y. Mung, Chetwyn C. H. Chan

**Affiliations:** 1Department of Rehabilitation Sciences, The Hong Kong Polytechnic University, Hong Kong, China; 2Department of Psychology, The Education University of Hong Kong, Hong Kong, China; 3Department of Mechanical Engineering, The Hong Kong Polytechnic University, Hong Kong, China; 4Innovation Technology Company Limited, Hong Kong, China; 5Research and Development Office, The Education University of Hong Kong, Hong Kong, China

**Keywords:** SOR, ASD, underlying mechanism, interventions, sensory-motor processing

## Abstract

Individuals with autism spectrum disorder (ASD) often exhibit sensory over-responsivity (SOR), which is characterized by an overwhelmingly negative reaction to or avoidance of sensory stimulation. Despite the detrimental effects of SOR on people’s personal and social lives, the knowledge of and interventions for the issue remain limited. This paper collates and reviews studies on SOR and information on the potential for effective interventions for people with ASD. This review reveals evidence that SOR has a close relationship with anxiety, depression, insomnia, and family life impairment and an underlying mechanism related to SOR. Four interventions and their theoretical bases in sensory-motor processing are discussed in this paper, namely, physical activity (PA), sensory integration therapy (SIT), mindfulness-based cognitive therapy (MBCT), and cognitive behavioral therapy (CBT). These interventions focus on establishing coping strategies for regulating the emotional response to sensory information, and they have been found to be effective and to have the potential to help children with ASD reduce their SOR behaviors. This paper provides guidance for selecting appropriate interventions and for further investigation of more effective interventions in the future.

## 1. Introduction

Autism spectrum disorder (ASD) is a pervasive neuropsychiatric disorder characterized by two symptom clusters of (a) social interaction/social communication deficits and (b) restricted repetitive patterns of behavior, activities, or interests [1]. In addition to the two core symptom clusters, individuals with ASD frequently show associated symptoms or symptom clusters that can significantly increase their functional impairment. Sensory over-responsivity (SOR), a condition that frequently causes clinical impairment and is characterized by unusual and heightened reactivity to sensations [2], is one of these symptom clusters. SOR is characterized by a faster, more intense, or longer response to sensation than typical sensory responsivity [3], and it occurs more frequently among children with ASD [4] than among typically developing children [5]. 

Professionals in a variety of fields have become increasingly aware of the sensory features of ASD over the past two decades [6,7]. In fact, the characteristic of “unusual interest in sensory aspects of the environment” was identified as a feature of ASD in the Diagnostic and Statistical Manual of Mental Disorders, 4th edition (DSM-IV). In 2013, the Diagnostic and Statistical Manual of Mental Disorders, 5th edition (DSM-5), added both sensory hypo-reactivity and hyper-reactivity to the diagnostic criteria as core symptoms of ASD [1]. Sensory features can be categorized into three patterns, namely, SOR (also known as sensory hyper-reactivity), sensory under-responsivity (SUR, also known as sensory hypo-reactivity), and sensation seeking (unusual sensory interests) [3], and many individuals with ASD show more than one sensory pattern. The extent of unusual sensory features has been found to be associated with the severity of autistic traits, such as repetitive behavior [8,9] and stereotyped behavior [9,10,11,12]. SUR refers to a lack of awareness of certain stimuli or a slowed response to sensory inputs, such as sounds or spoken language [13,14]. Conversely, SOR refers to the subjective experience of sensory overload that would not be regarded as troublesome to those without SOR [12,14,15]. Dunn’s model of sensory processing identified SOR as a low neurological threshold and a passive self-regulation strategy in response to sensory stimuli [16]. SOR has been reported in several studies to be more prevalent than SUR, with a prevalence of 56% to 79% in patients with ASD [4,17,18], and to affect almost all sensory modalities [6,19,20]. 

SOR is a condition in which exaggerated or prolonged responses to sensory stimuli result in interference with an individual’s participation in daily life [21]. Although SOR has the highest prevalence in ASD, it also appears in healthy people (8–28%) [22] and other diseases such as ADHD (about 46%) [23]. The existing criteria for defining SOR have been criticized for not being specific enough for the identification of the abnormalities [24]. First, SOR has not been formally included as a diagnosis in the DSM-5 or the International and Statistical Classification of Diseases and Related Health Problems [25]. Second, SOR has been associated with other psychopathology [26] or anxious/disruptive behaviors [27]. For example, one study assessed the sensory and anxiety symptoms of 191 typically developing children from a local community. Forty-three percent of preschool children with SOR also had a concurrent impairing anxiety disorder, and this was found after parents reported several additional behavioral challenges previously found to be associated with both sensory over-responsivity and anxiety separately: irritability, food selectivity, sleep problems, and gastrointestinal problems. The symptoms of sensory over-responsivity in preschool were found to be a strong and positive predictor of anxiety symptoms in children aged six. As a one-way connection, this was a rather specialized one. Finally, the symptoms of anxiety experienced during school years moderated the connection between sensory over-responsivity experienced during preschool years and both irritability and sleep issues experienced during school years. This research indicates that heightened sensitivity to sensory input is associated with an increased chance of developing an anxiety condition. Preschoolers who exhibit signs of sensory over-responsivity are more likely to show symptoms of anxiety as they enter elementary school, and these symptoms are linked to more behavioral difficulties later on [28]. Similarly, the co-occurrence of SOR and obsessive-compulsive symptoms in childhood and early adolescence was also found. SOR symptoms are highly associated with concomitant obsessive-compulsive symptoms [29]. Avoidant behavior and complaints of patients with SOR and obsessive-compulsive symptoms not being able to ignore and habituate to the sensory environment may be attributable to a cognitive bias of prediction [30], which may be explained by the habituation process. Based on these findings, it is reasonable to assume that psychoeducation and intervention strategies that encourage participants to draw on stored sensory memories can help people with obsessive-compulsive symptoms and SOR lead more productive lives.

There are two SOR subtypes or two kinds of symptoms that are relevant to ASD: (1) internalizing symptoms, namely, sensitivity, responding with distress, fear, and distractibility, and (2) externalizing symptoms, namely, avoidance, responding with withdrawal, defiance, resistance, and taking control over input [21]. The Interdisciplinary Council of Developmental and Learning Disorders [31] described the two subtypes in children under 3 years of age as a “fearful and anxious” subtype versus a “negative and stubborn” subtype. There is preliminary evidence to support these subclassifications [32]. SOR is also notably linked to higher impairment and distress, including more severe autism symptoms and internalizing symptoms such as anxiety [33,34]. Some studies have found that sensory symptoms and repetitive behaviors frequently co-occurred in children and adolescents with ASD (aged 6–17 years old) by parent report measures [35,36,37], and these are also associated with heightened emotional symptoms [38,39], particularly anxiety [40,41]. Furthermore, sensory complaints have been demonstrated to interact with attention skills [39,42]. In particular, high levels of SOR behavioral symptoms predicted high levels of repetitive behaviors, and the relationship remained the same when controlling for mental age [37,43]. This significant correlation was not found in other sensory abnormalities (sensory seeking and sensory under-responsivity) [43]. Restricted and repetitive behaviors also predicted the severity of anxiety symptoms in ASD [44,45], and SOR in ASD positively correlates with the severity of internalizing problem behaviors/or depression and anxiety [46,47,48]. 

In addition to ASD, SOR symptoms are also found in other types of developmental and psychopathological disorders, such as attention deficit hyperactivity disorder [49], fragile X chromosome [50], obsessive–compulsive disorder [51], and anxiety disorders [52]. It was also found that SOR may contribute to increased anxiety in children with intellectual disability and ASD, and lower-order repetitive behaviors attempt to alleviate that anxiety [53]. Questions about their different performances in ASD, both in terms of scale and nature, are raised. It is not clear to what extent SOR symptoms are related to the level of development. There is evidence that children and adults with ASD have higher intellectual abilities and fewer sensory abnormalities [6]. There is no direct evidence to examine the relationship between SOR and cognitive level in ASD; however, sensory alterations have been found to be related to cognitive level in developmental age in children with ASD, although the idea is still under investigation and has no consistent result. The studies of Baranek et al. (2006) and Narzisi et al. (2022) found that cognitive abilities were associated with the severity of sensory alterations [4,54]; however, an earlier study [55] failed to replicate the same results when matching groups according to their mental age, which relates to individual cognitive ability. More studies still are advocated to test the hypothesis by matching participants according to their cognitive levels. SOR symptoms have been associated with impairment in a wide range of areas, including reduced social functioning and adaptive skills in children with ASD [42,56,57]; social skill deficits [39]; and a higher incidence of internalizing symptoms, such as increased anxiety and depression [38,48], as well as externalizing behaviors, such as an increase in insomnia and disruptive behaviors [27,58]. SOR symptoms have also been found to contribute to family life impairment and maternal affective symptoms, increasing maternal stress and the disruption of family life routines [59].

Currently, the available methods of examining sensory processing include questionnaires, direct assessment, and physiological and neurological response measurements [7,60,61], but parents or caregiver reporting and observational methods are most commonly used for SOR in ASD as standardized assessments [55,62]. Of the few existing assessments, most are for children, and there are few standardized observational assessments specifically tailored for adolescents and adults [7]. These instruments measure SOR by assessing the occurrence or intensity of behavioral and emotional responses of individuals towards sensory stimuli across sensory domains (e.g., auditory, visual, tactile, vestibular, proprioceptive, gustatory, and olfactory sensations) (e.g., negative emotional reactions to sensory stimuli) [63]. The more frequent or intense the responses, the more severe the individual’s SOR symptoms [21,63]. However, there is conflicting evidence regarding the sensory symptom process of SOR in ASD across the lifespan. Some studies have found reduced sensory symptoms in adolescents and adults compared with children with ASD [64,65], and some studies have shown that symptoms increase with age [66], while others show stability throughout childhood [55,67]. The symptom development of SOR was also found to be a non-linear process, with the 6–9-year-old age group showing more severe SOR symptoms than younger and older age groups. So, recent evidence is valuable for examining different age-related changes. SOR symptoms decreased with age due to neurohormonal changes and/or the establishment of coping strategies [6,68]. A parent or caregiver reporting assessment has the advantage in that it may provide cross-context information from someone very familiar with the child, allowing researchers to collect such information more quickly and easily [60]. However, this method is too subjective, and parents or caregivers may give answers according to the researchers’ expectations. Direct assessment (observation) in conjunction with a parent or caregiver report is currently the best approach, but it is time-consuming, so there is a need to develop standardized observational assessments of sensory characteristics that can be used across age groups, including adolescents [68]. These inconsistent findings across cognitive levels and ages, in addition to the heterogeneity present in ASD, may be attributed to a range of methodological challenges, including comparing different sensory modalities across studies, different survey-based and physiology-based measures, and changing scientific standards over time. There is a lack of knowledge and exploration of a more effective assessment for SOR, as well as of the relationship with age. 

To explain the SOR phenomenon, researchers have proposed theoretical frameworks, such as the “weak central coherence” theory [69], the “temporal binding deficit” hypothesis [70], and the “excitatory/inhibitory imbalance” model [71]. Regarding the nature of SOR, it has been conceptualized as one of the patterns of sensory modulation disorder under the taxonomy of sensory processing disorder [3]; a self-regulation technique in individuals with a low sensory threshold under Dunn’s model of sensory processing [16]; and the result of disruption of neural development, which is crucial for processing visual, auditory, and tactile information [72,73]. However, individuals with SOR have excellent detection abilities (owing to a low neurological threshold); therefore, they may notice numerous things in their environment. For example, some children with SOR to sounds may protect their ears or ask people to be quiet, but they may also have perfect pitch in music.

Although the specificity of and mechanisms underlying SOR in ASD are still under investigation, some achievements have been made in prenatal and genetic factors, excitatory/inhibitory neurochemical balance, and brain connectivity.

(1) Prenatal factors

Studies have revealed that prenatal and genetic factors are common in both ASD and its SOR-related symptoms. Babies who were diagnosed as preterm birth and received prolonged neonatal intensive care unit (NICU) care were found to have a high prevalence of SOR [74]. Infants and toddlers of shorter gestational age were found to manifest frequent SOR-related behaviors [75]. However, Keuler (2011) reviewed a wide spectrum of prenatal factors and did not find them to significantly contribute to SOR symptoms in toddlers [22]. However, other studies explained that the brain volumes of babies with ASD being smaller than that of typically developed babies would have resulted in the delayed cortical maturation [71] and subnormal neuroplasticity [76] of the premature and light-birth-weight babies contributing to SOR. Prenatal exposure to substances, such as medications, alcohol, and testosterone, was associated with increases in symptoms of SOR in both human [22] and animal models [76,77]. Changes in the internal environment during the prenatal period due to the intake of medication or alcohol and to the increasing hormones produced by excessive stress can disrupt the balance of the pregnancy environment and affect the normal development of the sensory nervous system [77,78]. Similarly, an increased secretion of testosterone in the mother during pregnancy was suggested to impact the normal development of the sensory nervous system [79].

(2) Genetic factors and excitatory/inhibitory neurochemical balance 

There has been plenty of studies that have linked heredity to the presence of SOR-related symptoms. Twin studies showed strong genetic influences on traits of SOR [80,81]. The mothers of children with ASD with SOR symptoms obtained significantly higher scores (more serious) on a scale of SOR than those of typically developed (TD) children [82,83]. Studies found that mutations of neuroligin-3 [84] and GABRB3 [85] were significantly associated with sensory abnormalities in individuals with ASD. Other studies reported results corroborating that these mutations upset the balance of gamma-aminobutyric acid (GABA) [86,87] and glutamate [88,89]. The imbalanced synthesis of GABA and glutamate resulted in excessive inhibitory or excitatory responses to sensory stimulations in children with ASD [71]. In particular, the upset of the synthesis of GABA was attributed to SOR. A recent MR spectroscopy study reported close relationships between the levels of GABA concentration in the cortex of the brain and the visual perceptual functions of children with ASD, and it was found that the participants with ASD had lower levels of GABA in the visual cortex and were more likely to have visual sensitivity [90]. A recent study in children with ASD examined the relationship between thalamic and soma-to-sensory cortex inhibitory (GABA) and excitatory (glutamate) neurochemicals, with intrinsic functional connectivity, found that thalamic sensory gating, which may be associated with an excitatory/inhibitory (E/I) neurochemical imbalance, and SOR severity were negatively correlated with thalamic GABA and positively correlated with somatosensory glutamate [91]. These results imply that disturbances in the thalamic neurochemical balance may interfere with the thalamus’s involvement in integrating, transmitting, and suppressing attention to sensory input. These findings have implications for future studies and pharmacological therapies that modulate GABA.

(3) Brain connectivity

Studies have demonstrated that SOR in ASD is related to structural and functional brain connection impairments. First, there is decreased inter-hemispheric structural covariance connectivity among subcortical regions and increased intra-hemispheric structural covariance connectivity, which produces symptoms of weak central coherence [92], and there is increased structural covariance in the right cerebral hemisphere, which is linked to the theory of hemispheric functional lateralization [93]. Second, there is decreased functional connectivity between the thalamus and cortex, and there is increased functional connectivity between the thalamus, sensory cortex, and amygdala, resulting in reduced top-down regulation from the cortex to the thalamus; this causes difficulty in filtering out and/or integrating sensory information, as well as failure in selective inhibition and attention to external stimuli, resulting in decreased inhibition in generating excessive responses to the incoming sensory stimuli [94]. This prevents adequate regulation and integration of sensory information in the brain, leading to the output of excessive emotional responses and, consequently, more severe SOR symptoms in ASD. Thus, the possible neural mechanism underlying SOR in ASD can be summarized as follows: the abnormalities of functional and structural connectivity in the brain of individuals with ASD inhibit proper top-down regulation and the integration of sensory inputs, leading to increased attention to extraneous sensory stimuli during sensory processing and, as a result, the output of excessive emotional responses to these stimuli. These findings suggest that effective interventions can be designed to change one of the processes of SOR by strengthening or weakening functional connectivity. Figure 1 shows a summary of background knowledge related to SOR.

Although there are interventions developed to address SOR in ASD, more research is required to deconstruct the specific systems that may be affected by the interventions. In this paper, we select and summarize potentially effective interventions for SOR, which may tackle the underlying mechanism of SOR.

## 2. Method

### 2.1. Aims of the Current Review

A few reviews have been published on sensory alteration in general [6,95], summarizing SOR symptoms and impactors, such as age, IQ, gender, and assessment methods. However, explanatory factors of the core nature of ASD with SOR are yet to be revealed. Although many effective interventions targeting core ASD symptoms (social skills and repetitive behaviors) [96] and a few studies on the intervention for sensory alteration in ASD have been found, little is known about effective interventions for SOR. Therefore, based on previous studies, we aim to answer the following question through this review:

Research question: As a core feature of ASD, can effective interventions be developed for SOR in isolation based on the existing theoretical basis and empirically supported sensory abnormality therapy?

### 2.2. Search Strategy

In this review, we found and analyzed papers using a hybrid methodology. Firstly, we identified relevant studies by performing a systematic literature database search. Keywords (see Table 1) were entered into PubMed, Web of Science, and Google scholar to find relevant English-only published papers. Then, we summarized and evaluated the findings of the review after conducting a narrative review to identify prominent themes in the literature. We did not conduct a qualitative literature synthesis.

### 2.3. Study Selection

Before beginning the literature search, inclusion and exclusion criteria were developed. The search was limited to quantitative intervention studies on ASD and SOR, whereas case studies were eliminated. Four types of effective interventions with potential neural mechanisms in sensory alteration were included: physical activity (PA), sensory integration therapy (SIT), mindfulness-based cognitive therapy (MBCT), and cognitive behavioral therapy (CBT). The electronic search was concluded, and duplicates were eliminated. It is important to note that we initially selected sensory interventions for ASD. However, in the ASD group, there was no sensory MBCT intervention research, only studies on other ASD symptoms (e.g., anxiety). Thus, we picked an MBCT study connected to sensory intervention for individuals with SOR in order to offer readers information regarding the viability of this intervention for the ASD population. This review only selected six studies of successful sensory interventions: one in PA, three in SIT, one in MBCT (individuals with SOR), and one in CBT (see Table 2 ). In this review, we do not explain additional research details but rather the brain mechanism’s theoretical foundation in order to examine the efficacy and practicability of future interventions for SOR in ASD.

## 3. Result

### 3.1. Interventions for SOR in People with ASD

Although there is a lack of neuroscience to measure the behavioral intervention response in ASD with SOR, we are still able to propose a hypothesis suggesting that effective intervention is more likely to change the behavioral response by changing the processing of sensory information or neural structure. Exploring the neural correlates of behavioral interventions meant to address the primary symptoms of ASD has shown some success. In a review study, Stavropoulos summarized four studies that measured brain activity using functional magnetic resonance imaging (fMRI) or electrophysiology, both before and after behavioral intervention, and found that decreased core symptoms of ASD after behavioral intervention were associated with changes in brain activity [103]. Baker et al. (2020) also found in an event-related potential (ERP) study that decreased social communication deficits and improvements in social skills in adolescents with ASD result in behavior-related brain activity before and after behavioral intervention [104]. Considering the high prevalence of SOR and the seriousness of its effects in children with ASD, appropriate interventions are particularly important and should address underlying neural activity issues. While little is known about the neural basis for current behavioral interventions for SOR, reviewing the three major aspects of sensory–motor processing (sensory, emotional, and motor components) [105] might offer some insight into choosing effective interventions (Figure 2). 

Sensory–motor processes mostly occur in working memory, making it possible for reasoning and decision making to occur through the top-down regulation of attention resources [106,107] before motor responses are completed. Taking into account the changes in sensory–motor processing in children who have ASD with SOR, this paper identified four types of interventions, namely, PA, SIT, MBCT, and CBT, that may benefit this population. Emphasis is placed on three aspects of these interventions: their theoretical rationale, their possible neural basis, and their potential value for managing SOR in children with ASD.

### 3.2. Physical Activity (PA)

PA is often regarded as a polypill. Early research found that neurotypical individuals with regular PA demonstrate a reduced auditory hearing threshold of 4 to 6 decibels in high-frequency sounds (i.e., 2000 to 4000 Hz) [108]; recent findings have indicated that regular PA is negatively correlated with sensory sensitivity and emotional disturbances in neurotypical individuals [109]. Individuals with SOR may have a low perceptual threshold for stimuli, but a 12-week horseback training program was revealed to significantly reduce sensory sensitivity among children with ASD when compared with those in a waitlist control group [97]. Regarding the mechanisms underlying PA, one study revealed that regular PA resulted in structural changes to the anterior cingulate cortex (ACC), related to a reduction in negative emotions [110]. Lin et al. (2020) recruited adolescents aged 12–14 with symptoms of depression and randomly assigned them to a psychoeducation control group or an aerobic exercise intervention; when comparing structural changes in the ACC from baseline to post-intervention in the MRI data, Lin found that the participants in the exercise group displayed increased gray matter volume of the left rostral ACC, increased cortical thickness of the right rostral ACC, and alleviation of mood symptoms when compared with the control group. Specific to sensory processing sensitivity, it was speculated that PA would serve as a moderator that decouples sensory processing from emotional arousal [111]. Evidence of the effect of PA on SOR is still scarce and merits further study.

### 3.3. Ayres Sensory Integration Therapy (SIT)

Ayres (1972) proposed the original theoretical underpinning of SIT, which emphasizes the importance of active and dynamic sensory–motor processing for supporting functional development and interaction within social and physical environments [23]. A typical SIT has three components: the registration of sensation in the central nervous system; modulation to regulate inhibition or propagation of neural signaling; and motivation to use motor skills to respond to or ignore the registered sensory stimuli with the motor ability [112]. The integrating processes may involve plastic changes in neurons, the diencephalon, and within and between hemispheres [23,113]. This intervention is typically utilized by physical and occupational therapists [112] and falls outside of the jurisdiction of mental health specialists. This treatment aims to incorporate sensory-relevant activities into play and adaptive behaviors, therefore improving the child’s ability to process sensory information [114]. 

Under ASI, SOR in people with ASD is considered a “disorder of modulation” [109]. The problems manifest as a failure to detect and interpret sensory inputs in the registration stage, subsequently inhibiting and propagating registered inputs in the modulation stage. As such, the purpose of ASI is to enhance the sensory–motor processes of registration and modulation. Evidence has accumulated indicating that ASI can be an effective intervention for SOR in people with ASD [98,99,100].

Recent studies on ASI have suggested that ASD with SOR is related to abnormal changes in a few key neural substrates, such as the posterior cingulate cortex, the insula, the thalamus, and the amygdala [23,109]. When abnormalities are seen in these neural substrates, ASI includes the examination of autonomic nervous system responses to sensation; the role of arousal in attention allocation; and integration among the sensory systems in the service of functions, such as self-regulation, focusing, and the shifting of attention [23,115,116]. This mechanism of sensory integration as an explanation for SOR shares common ground with what this paper proposes. Further research into how ASI taps into sensory–motor processes is highly recommended.

### 3.4. Mindfulness-Based Training (MBCT)

MBCT has been proposed as useful for tackling psychological distress in individuals with SOR [117]. Mindfulness was defined by Kabat-Zinn (2009) as being characterized by paying attention to the present moment with a receptive and non-judgmental attitude [118]. MBCT is a psychosocial intervention that combines cognitive behavioral techniques and mindfulness strategies to manage cognitive reactivity and emotional regulation [119]. Studies have reported the effectiveness of mindfulness in mediating sensory processing in individuals with sensory processing sensitivity [118], as well as in individuals who struggle with emotional regulation [120,121]. Other studies have reported that mindfulness practice is effective in enhancing perceptual sensitivity, maintaining sustained attention [122], and improving focus [123]. Prior research has shown that mindfulness is beneficial in reducing a number of mental health issues in a wide variety of populations. Through a comprehensive review, Cachia et al. (2016) revealed the effectiveness of mindfulness therapies in lowering stress, anxiety, sadness, rumination, and aggressiveness, while enhancing the positive impact and mental health in individuals with ASD [124]. The value of mindfulness for SOR is in enhancing attention to incoming sensory stimuli and improving emotional arousal towards these stimuli. 

Mindfulness training has been shown to result in increasing deactivation of the amygdala [125]. Such neural effects have prompted researchers to suggest that this training has the potential to be useful in enhancing self-control and reducing emotional reactivity in individuals with SOR [126]. It is important to note that no published studies have been found on the effects of mindfulness training on SOR in individuals with ASD. Further research is needed to fill this gap and to explore the clinical effects of mindfulness training on SOR and the mechanisms behind such effects. 

### 3.5. Cognitive Behavioral Therapy (CBT)

CBT makes use of the interplay between an individual’s thoughts, feelings, and behaviors [127]. As with mindfulness training, there have been a handful of studies on CBT in relation to people with ASD but not necessarily in relation to people with SOR. For example, Maddox et al. (2017) revealed that CBT can improve social skills and anxiety in the long term in adolescents with ASD [128], and for children with ASD, CBT for anxiety disorders is also effective [129,130]. For young high-functioning children with ASD, CBT may improve the regulation of anger and anxiety [131]. Similarly, CBT may also reduce anxiety symptoms in youth with ASD aged 8–14 years old [132,133]. Eighteen cognitively able adult males with ASD improved their social anxiety after CBT [134]. The most relevant study investigated people with ASD with sensory processing difficulties using a two-part intervention: one part involved deliberate focus on the sensations, while the other part involved strategies for coping with the sensory experience [102]. The participants were able to practice self-regulatory techniques to modify their behaviors and to alleviate the unpleasantness associated with sensory stimuli. However, no study has been found on CBT as a clinical intervention for individuals with ASD and SOR. Although CBT targets emotion and response, it is not designed to specifically address sensory impairments. Researchers identified and proposed three potential changes to CBT in order to treat SOR in children and adolescents with ASD, and these changes seek to directly address the potential mechanisms behind the development of sensory abnormalities [135], but the effect has not been verified. It has been suggested that combining CBT with MBCT is useful for modulating symptoms and reducing psychological distress for individuals who have ASD and SOR [117]. The MBCT program tries to improve one’s response to an event by changing negative thought processes via the practice of acceptance and awareness that is free of judgment. The rationale behind this proposition is that the combined intervention would moderate the bi-directional stimulation of bottom-up (sensory-perceptual) sensitivity and top-down (cognitive) reactivity, making room for the choice of an appropriate response. 

The processing of incoming sensory signals triggers cognitive and emotional processes, in which the hippocampus and amygdala should both be involved. At the same time, emotions impact the cognitive process, in turn feeding back into and modifying both sensory and emotional processes. Negative emotions induce cognitive reactivity, which, in turn, feeds back into negative emotions, resulting in the formation of two vicious loops. Circle 1 involves emotional processing, which can cause emotional distress, such as depression and worry, whereas circle 2 involves sensory cognitive processing, which causes somatic distress. In turn, these vicious loops may have harmful effects on behavior. According to the proposed paradigm of MBCT, it is neither the strong sensory input itself nor the associated negative emotions that induce psychological discomfort in individuals with SOR but rather their following cognitive reactions. Future research is needed to gain a deeper understanding of how CBT can tap into both the attention and emotion components of SOR in individuals with ASD. 

## 4. Discussion

In summary, the implications of these findings on interventions for people with ASD suggest that, instead of normalizing sensory processing, the focus should be on developing coping mechanisms for managing attention and emotional responses to stimuli [136,137]. Existing effective interventions include the teaching of coping methods to lessen the anxiety in individuals with ASD [138,139]. Given the substantial co-occurrence of anxiety and SOR in children with ASD, these therapies may be adaptable to target SOR [140]. PA, SIT, MBCT, and CBT have the potential to foster children with ASD in developing new emotional regulation methods in response to sensory stimuli (see Figure 2). A recent study identified the importance of the interplay between physiological regulation, cognitive abilities, and behavioral regulation when individuals encounter or participate in sensory activities [141]. For these four kinds of interventions, many factors may contribute to the therapeutic outcomes, such as the duration of the intervention and the age and cognitive ability of the individual. However, it is not clear how these factors affect the process of remediating the sensory over-responsivity of individuals. Due to the heterogeneity of individuals with ASD, further investigation of the moderating and mediating effects of the emotional regulation, physiological regulation, and cognitive ability of individuals on these interventions is recommended. This would help identify the underlying mechanisms of SOR and allow for the design of more effective interventions for people with ASD and SOR.

These findings suggest that SOR might share neurobiological mechanisms with repetitive behaviors and anxiety. Clear mechanisms are important for both diagnostic classification and intervention. Most interventions address the first two core symptoms of ASD, and only one–three studies per intervention, in which subjects were either children (5–12 years old) or adolescents (11–16 years old), specifically address the sensory symptoms. Due to the limited number of studies searched, we are currently unable to disaggregate and compare subjects by age in these studies. There are still other issues to be resolved. Existing diagnostic approaches, for instance, are heterogeneous and lack clear measurement standards for various age groups. The current research on the association between SOR and age, cognition, and other characteristics, as well as the neural mechanism underlying SOR, is inconclusive. These issues are also important as to whether effective interventions specifically targeting SOR can be developed.

## Figures and Tables

**Figure 1 children-09-01584-f001:**
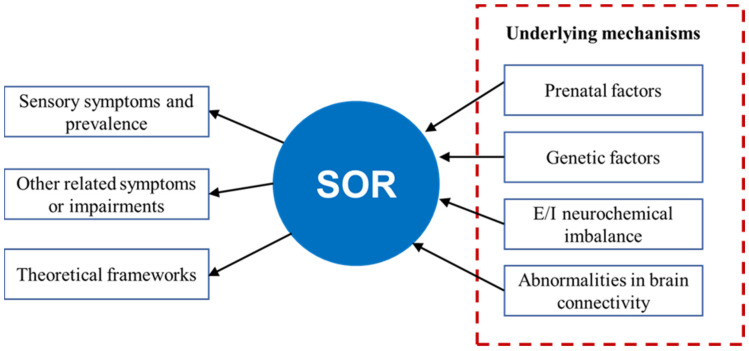
Summary of SOR, namely, symptoms, prevalence, co-occurrence symptoms, theoretical frameworks, and underlying mechanisms.

**Figure 2 children-09-01584-f002:**
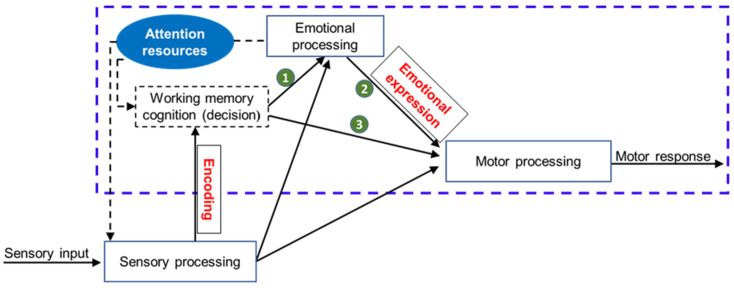
Summary of sensory–motor processing with sensory, emotional, and motor components, from detection of sensory input to response. ***Note:*** One theory views SOR from the perspective of sensory–motor processing. Sensory inputs (sensory signals) first enter sensory processing for registration, detection, and encoding. Next, the signals enter emotional processing for emotional expression (e.g., negative emotions, such as crying) or enter motor processing (e.g., they elicit a motor response to express an emotion or respond to sensory signals) based on previous memory or cognition. To help children adapt to their environment, an intervention involving top-down regulation (e.g., bringing attention to or away from these signals—the box outlined in blue dashes) may lead to the selection of an appropriate behavioral response (e.g., emotional expression or motor response). Such intervention may elicit responses in various configurations: the cognition–emotion process (green number 1; PA, CMBT), the emotion–motor process (green number 2; PA, SIT, MBT, CMBT), and the cognition–motor process (green number 3; SIT, CMBT). CMBT = cognitive-behavioral therapy; MBT = mindfulness-based training; PA = physical activity; SIT = sensory integration therapy.

**Table 1 children-09-01584-t001:** Search terms in studies of interventions.

Population Term	Sensory Term	Descriptor Term	Intervention Types
Sensory hyper-responsivity OR hyper-responsivity OR Over-responsivity OR hyper-reactivity	Autism OR autistic syndrome disorder OR autistic disorder OR autism spectrum disorder OR Asperger	Effective intervention OR training OR therapy	PA, SIT, MBCT, and CBT

**Table 2 children-09-01584-t002:** Six sensory studies related to proposed effective intervention for SOR in this review.

Authors	Number/Age of Participants	Assessment	Intervention/Duration	Metrics of Sensory Alteration	Sensory Processing Results
Bass et al. (2009) [97]	19 ASD in experimental group 15 ASD as control group (Age: 5–10 years old)	Autistic traits: DSM-IV-TR autism spectrum diagnosis by doctor Sensory or SOR symptoms: Sensory Profile (SP) by parents or teachers	PA: 1 h/wk; 12 weeks	The score of specific subscales of SP of sensory sensitivity was used to assess sensory alteration pre- and post-intervention	Experimental group demonstrated significantly improved social motivation and sensory sensitivity, as well as decreased inattention and distractibility
Fazlioğlu et al. (2008) [98]	15 children with ASD in the experimental group 15 children with ASD in the control group (Age: 7–11 years old; low functioning)	Autistic traits: DSM-IV autism spectrum diagnosis by doctor Sensory or SOR: sensory evaluation form by parents/caregivers and researcher	SIT: 45 min/wk; 12 weeks	The scores of the evaluation forms were reassessed and compared between pre- and post-test	Sensory problems of children with autism improved, including hyper- and hypo-sensitivity to stimulation in experimental group
Kashefimehr et al. (2018) [99]	16 children with ASD in the experimental group 15 children with ASD in the control group (Age: 3–8 years old)	Autistic traits: DSM-5 Sensory or SOR: two or more subscales of the SP by parents/caregivers	SIT: 1 h/wk; 12 weeks	The pretest and posttest SP scores were compared between the intervention	The intervention group showed significantly greater improvements in all factors and domains of SP
Pfeiffer et al. (2011) [100]	18 children with ASD in intervention groups (Age: 6–12 years old)	Autistic traits: DSM-IV Sensory or SOR: Sensory Processing Measure (SPM) questionnaire by researchers and parents or caregivers	SIT: 45 min/wk; 6 wks	The pretest and posttest SPM scores were compared between the intervention	No significant differences in the scores on the SPM. A significant decrease in autistic mannerisms.
Bakker et al. (2012) [101]	111 highly sensitive people (79 females, 32 males) (Age: 18–64 years old)	Autistic traits: non-autistic person Sensory or SOR: Highly Sensitive Person Scale (HSPS): a self-report measure of Sensory-Processing Sensitivity Mindfulness awareness: The Mindfulness Attention Awareness Scale Other scale: The Depression, Anxiety and Stress Scales All surveys were completed by self-report.	MBCT: no intervention, only tested mindfulness awareness	In this research, we aimed to investigate the relationship between SPS, mindfulness, and distress using a cross-sectional methodology in a non-clinical sample	SPS was related to higher levels of depression, anxiety, and stress. Participants with low acceptance of mindfulness in particular had higher anxiety, while the SPS score of those with high acceptance of mindfulness had no significant relationship with anxiety.
Edgington et al. (2016) [102]	7 cognitively able adolescents with ASD (Age: 11–16 years old)	Autistic traits: ICD-10 or DSM-IV-TR criteria, an independent clinical diagnosis of ASD or Asperger’s Syndrome IQ: Wechsler Abbreviated Scale of Intelligence-II; (WASI-II; 70 or above) Sensory or SOR: Adolescent/Adult Sensory Profile (AASP) self-report	CBT: 45 min/wk; 8 wks	The pretest and posttest AASP scores were compared between the intervention	The intervention itself was feasible.No significant difference between pretest and posttest AASP scores.Adolescents with autism reported feeling better able to deal with sensory issues.Effectively raised meta-conscious awareness and self-regulation.

## Data Availability

All data generated or analyzed during this study are included in this article. Further enquiries can be directed to the corresponding author.

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
