# Peer review of "Interventions for Sensory Over-Responsivity in Individuals with Autism Spectrum Disorder: A Narrative Review"

_children, 2022, doi:10.3390/children9101584_

Round 1

Reviewer 1 Report

Dear Editor, thank you for the opportunity to review the paper entitled “Interventions for Sensory Over-Responsivity in Individuals with Autism Spectrum Disorder ASD”.

This work tries to summarize the state-of-the-art in one important subfield of Autism. I believe that focusing on sensory over-responsivity in the context of Autism intervention may be relevant for treatment outcome and to better understand the different dimensions that are relevant for the response and that impact the quality of life of autistic people.

I will outline some general comments and areas that I believe should be better addressed in the manuscript to improve its quality and to give a more comprehensive view of the topic before considering it for publication on “Children”.

Introduction

-       The DSM-5 specifies two (not three) symptom clusters for ASD: (A) social communication/social interaction deficits and (B) Restricted repetitive patterns of behavior, interests, or activities.

-       The introduction covers well the definition, the phenotypes, and the impact of sensory alterations, focusing on SOR. However, the discussion is focused on general outcomes and psychiatric comorbidities. Since Autism has different known comorbidities also with other neurodevelopmental disorders, and it is often associated with cognitive impairment i suggest to include these conditions in the introduction, citing relevant literature with respect to the possible association between cognitive functioning and sensory alterations.

-       I would suggest as well to include a discussion with respect to the possible association with sensory alterations and the severity of symptoms of cluster B.

-       Which are (and what type of) instruments are available to measure sensory alterations/SOR? Which relationships with gold standard clinical measures? This aspect is not addressed and could be important to better understand the available literature and the specificity of sensory aspects.

-       Lines 138-140: I suggest rephrasing since this sentence is not that clear.

-       In the second part of the introduction I would suggest to include some literature about hypotheses and evidence that link behavioral and neural phenotypes, also in relation to intervention. See for example the following paper that contains an interesting discussion about the complementarity between the social motivation hypothesis and the intense world hypothesis with respect to neural phenotypes and intervention effects

-       Stavropoulos, K. K. M. (2017). Using neuroscience as an outcome measure for behavioral interventions in Autism spectrum disorders (ASD): A review. Research in Autism Spectrum Disorders, 35, 62-73.

-       I would also suggest to widen the discussion including the relation between age and sensory alterations, possibly with literature on developmental trajectories in sensory alteration. Are there age-related differences? Do sensory alterations/SOR changes over time? This could be helpful while reflecting on the possibility that different interventions may show different efficacies based on age. For example:

-       Chen, Y.-J., Sideris, J., Watson, L. R., Crais, E. R., & Baranek, G. T. (2022). Developmental trajectories of sensory patterns from infancy to school age in a community sample and associations with autistic traits. Child Development, 93, e446– e459. https://doi.org/10.1111/cdev.13745

-       It is unclear how the authors selected the types of interventions and especially how they selected the literature to discuss. Was some systematic criteria used? Or is this a narrative review with literature selected by the authors without rigorous criteria? This part is missing and I think it is fundamental and it should also be specified in the title. Also, parenthesis are missing in the title for (ASD).

Part 2

I find this part a bit lacking in structure and difficult to follow. I suggest to better organize the exposition and in particular:

-       Study results are reported in a narrative way but often details about participants are missing, and represent important information. It is important to specify which studies are related to individuals with ASD, with ASD and other psychiatric conditions, without ASD but with other psychiatric conditions, and TD/neurotypical. It is also important to specify sample sizes, the age of the participants, the study design, and at least the type of outcome measure. It is also fundamental to include some information about the effect sizes of the differences emerging in literature, when available, or report the lack of strength in the literature to guide future research. In studies with autistic subjects it could be important to include possible relationships between sensory alterations and the clinical measures of the sample, when available. I suggest adding a table with this information to increase the quality of the manuscript and its readability, and to critically discuss these aspects during the exposition, in particular with respect to the specificity of these interventions. In general, I suggest better structuring this part with a more clear exposition.

-       Studies about mindfulness and ASD are available, just an example: Cachia, R. L., Anderson, A., & Moore, D. W. (2016). Mindfulness in individuals with autism spectrum disorder: A systematic review and narrative analysis. Review Journal of Autism and Developmental Disorders, 3(2), 165-178. As mentioned before, I would suggest to include a discussion with the relationship between sensory alteration and symptom severity since it could help in interpreting results from the clinical literature. For example, in the ADOS-2 the CRR score includes aspects of sensory alterations and may be available as clinical measure in several studies.

-       Studies on CBT for autistic individuals are available and could be discussed with respect to sensory alterations. The last part of the CBT section has no supporting literature that should be added.

-       For each type of intervention I would also suggest to include a discussion about the suitability for different age ranges, and the availability or lack of specific literature.

-       The review widely discusses the aspects of sensory alterations and in particular sensory over-responsivity, aiming at improving knowledge about specific interventions. However, there is no reference to any of the available, evidence-based interventions for ASD. Just an example among many other: Sandbank, M., Bottema-Beutel, K., Crowley, S., Cassidy, M., Dunham, K., Feldman, J. I., Crank, J., Albarran, S. A., Raj, S., Mahbub, P., & Woynaroski, T. G. (2020a). Project AIM: Autism intervention meta-analysis for studies of young children. Psychological bulletin, 146(1), 1–29. https://doi.org/10.1037/bul0000215. Sensory alterations are often targeted in different models of intervention, as well as emotional regulation. I consider it important to add a discussion about available evidence, relationship between intervention programs and sensory alterations, especially SOR, highlighting possible gaps in literature, as well as the possibility/feasibility to target and integrate sensory alterations/SOR into treatment models, since these represent the current golden standard for Autism treatment.

In the conclusion, I would better discuss the implications for clinical practice and the feasibility of integrating sensory alterations/SOR specific activities into intervention programs, also in terms of public health, costs, and potential impact on life quality. Finally, I would suggest better highlighting current research gaps, needs, and priorities to guide future research.

Reviewer 2 Report

This paper presents sensory over-responsivity, a prevalent   comorbid condition in children with autism spectrum disorder which is linked to anxiety, depression, insomnia, and family life impairment in this population. The paper describes the significance of the problem, possible mechanisms involved and behavioral-cognitive approaches to therapy.   While the presented materials encompass the main aspects of sensory over-responsivity in this population in a descriptive manner, metrics of the impairment and the therapeutic improvements is completely lacking. It is important to get a clear understanding what is considered to be over-responsivity, severity of the problem in an individual, how much improvement can be expected with each of the approaches presented, how far is the achievable response from the therapeutic targe and what is considered a clinically important response?  

Round 2

Reviewer 1 Report

The authors addressed all the comments.